# Data Augmentation for Motor Imagery Signal Classification Based on a Hybrid Neural Network

**DOI:** 10.3390/s20164485

**Published:** 2020-08-11

**Authors:** Kai Zhang, Guanghua Xu, Zezhen Han, Kaiquan Ma, Xiaowei Zheng, Longting Chen, Nan Duan, Sicong Zhang

**Affiliations:** 1School of Mechanical Engineering, Xi’an Jiaotong University, Xi’an 710049, China; zhangkai0912@stu.xjtu.edu.cn (K.Z.); hanzehen@stu.xjtu.edu.cn (Z.H.); mkq1994@stu.xjtu.edu.cn (K.M.); hlydx1314@stu.xjtu.edu.cn (X.Z.); cltdevelop@stu.xjtu.edu.cn (L.C.); shenkong@stu.xjtu.edu.cn (N.D.); zhsicong@mail.xjtu.edu.cn (S.Z.); 2State Key Laboratory for Manufacturing Systems Engineering, Xi’an Jiaotong University, Xi’an 710049, China

**Keywords:** motor imagery, CNN, DCGAN, data augmentation, classification

## Abstract

As an important paradigm of spontaneous brain-computer interfaces (BCIs), motor imagery (MI) has been widely used in the fields of neurological rehabilitation and robot control. Recently, researchers have proposed various methods for feature extraction and classification based on MI signals. The decoding model based on deep neural networks (DNNs) has attracted significant attention in the field of MI signal processing. Due to the strict requirements for subjects and experimental environments, it is difficult to collect large-scale and high-quality electroencephalogram (EEG) data. However, the performance of a deep learning model depends directly on the size of the datasets. Therefore, the decoding of MI-EEG signals based on a DNN has proven highly challenging in practice. Based on this, we investigated the performance of different data augmentation (DA) methods for the classification of MI data using a DNN. First, we transformed the time series signals into spectrogram images using a short-time Fourier transform (STFT). Then, we evaluated and compared the performance of different DA methods for this spectrogram data. Next, we developed a convolutional neural network (CNN) to classify the MI signals and compared the classification performance of after DA. The Fréchet inception distance (FID) was used to evaluate the quality of the generated data (GD) and the classification accuracy, and mean kappa values were used to explore the best CNN-DA method. In addition, analysis of variance (ANOVA) and paired *t*-tests were used to assess the significance of the results. The results showed that the deep convolutional generative adversarial network (DCGAN) provided better augmentation performance than traditional DA methods: geometric transformation (GT), autoencoder (AE), and variational autoencoder (VAE) (*p* < 0.01). Public datasets of the BCI competition IV (datasets 1 and 2b) were used to verify the classification performance. Improvements in the classification accuracies of 17% and 21% (*p* < 0.01) were observed after DA for the two datasets. In addition, the hybrid network CNN-DCGAN outperformed the other classification methods, with average kappa values of 0.564 and 0.677 for the two datasets.

## 1. Introduction

A brain-computer interface (BCI) is a communication method between a user and a computer that does not rely on the normal neural pathways of the brain and muscles [1]. Electroencephalogram (EEG) signals are widely used as a BCI input because the method is non-invasive, cheap, and convenient. The generation of EEG signals can be divided into two types: active induction, such as motor imagery (MI), and passive induction, such as steady-state visual evoked potential, P300, and auditory evoked potential [2].

MI is a mental process that imitates motor intention without real motion output [3], i.e., the brain imagines the entire movement without actually contracting the muscles. In the field of neurophysiology, there are many similarities between real movements and motor imagery because of the consistency of the peripheral autonomic nerves and the cortical potential [4,5]. Therefore, MI is a brain activity that is similar to real exercise and may cause a change in the potential of the cortex [6]. These actions are called event-related desynchronization (ERD) and event-related synchrony (ERS), which are used to distinguish features of different body movements [7]. As a result, the EEG signal can be decoded into different commands to control peripheral devices when the brain imagines different movements.

Traditional methods, such as machine learning and signal processing, are widely used in the study of signal processing for MI-EEG decoding [8,9,10,11,12]. In general, this process consists of the following steps: First, noise and irrelevant frequency bands are removed by preprocessing. Next, various mapping models are created for different categories of features to complete feature extraction. Finally, the different feature models are classified and decoded separately. Traditional methods require a multi-step process. If errors occur in the intermediate step, the processing results will be affected. The acquired EEG signals are extremely weak and are mixed with uncorrelated biological signals, resulting in challenges in the pattern recognition process. In addition, differences in the physiological structure across subjects/sessions will cause differences in the feature distribution of the MI-EEG signals. These problems greatly hinder the practical application of the BCI.

In recent years, deep neural networks (DNNs) have provided good results for the classification of linguistic features, images, sounds, and natural texts [13,14,15,16]. Due to their end-to-end model structure and automatic feature extraction ability, DNNs minimize the interference of redundant information and improve classification performance. The use of neural networks for MI-EEG signal decoding has several advantages. However, in practical applications, it is difficult to collect sufficient data due to the limitations of available subjects, experiment time, and operation complexity. The performance of DNNs is highly sensitive to the number of samples. A small-scale dataset tends to lead to poor generalizability during model training, which adversely affects the classification accuracy [17].

One promising approach to avoid overfitting and improve the performance of deep networks is data augmentation (DA) [18]. This technique augments data by artificially generating new samples based on existing training data [19]. Typical methods of DA include geometric transformation (GT), noise addition (NA) [20], and generative models [21,22,23]. DA using GT and NA is achieved by changing the geometric features of the data, and generative models use a hidden model to create generated data (GD) that have a similar distribution to the real data (RD) [24]. These DA methods increase the quantity and diversity of the original data, thus helping to understand the mathematical distribution of the original data. Examples of studies on DA are listed in Table 1.

As shown in Table 1, traditional DA approaches (GT and NA) show promise but the primary limitation of these methods is that they are unable to learn the statistical characteristic using raw data, especially for MI-EEG data, which exhibit strong randomness and non-stationarity [33]. The GT strategy essentially consists of increasing the number of training samples by applying a transformation, such as reflection, rotation, shear, and shift, to the training images. Shorten and Khoshgoftaar [34] mentioned that some methods of GT may destroy the data label that is strictly related to the location information. However, the MI-EEG features are closely related to the channels and frequency band, which may result in a confused response of the model for the output after the DA using GT. For example, if MI data are augmented using a rotation or shifting, we may alter the representation of the features [35]. NA, as another typical DA strategy, is achieved by adding random values drawn from a Gaussian distribution to the raw data. However, NA cannot effectively improve the diversity of the features and patterns of the data [36]. In application domains, such as biological signals and EEG signals, the biases distancing the training data from the testing data are more complex than the noise variances [37]. As for the EEG signal augmentation, generating realistic data requires a profound understanding of the morphology and patterns of the raw data. Especially for spontaneous potential signal-MI, the variability of features and patterns across subjects/sessions brought a huge challenge for creating a generated model that can effectively produce artificial signals that are similar to real signals. Therefore, it is necessary to explore an optimal DA strategy for the classification of MI-EEG. Traditional methods execute data augmentation based on the input space and ignore the probes for the feature space of data, while neural networks show an incredible ability regarding the feature extraction of data [38]. One of the methods used to augment feature space is an auto-encoder (AE), which maps raw data into low-dimensional data using an encoder and reconstructs these vectors back into an image using a decoder. However, the training process of AE is unstable and prone to produce meaningless results [39]. Recently, some studies have demonstrated that generative adversarial networks (GANs) are well suited for EEG-DA [25,40,41]. However, few studies were conducted on the analysis of MI signals.

In this study, we proposed a DA framework based on a deep convolutional generative adversarial network (DCGAN) to obtained spectrograms of MI data and a convolutional neural network (CNN) model to verify the classification performance after DA. We reviewed common DA methods for MI-EEG and compared the augmentation performance of these models based on the Freéchet inception distance (FID). Then, we combined these DA models with a CNN to classify the MI signals and evaluated their classification performance. The best DA model (CNN-DCGAN) was used, and its performance was compared with that of existing algorithms on a public dataset. The results show that the DCGAN was an effective DA strategy for MI-EEG; the proposed hybrid CNN-DCGAN model outperformed the best classification method in the existing literature.

The remainder of the paper is organized as follows. Section 2 describes the methods of the deep learning model and the generation of the artificial EEG signals. Section 3 presents the experimental results. The discussion follows in Section 4, and Section 5 is the conclusion of the paper.

## 2. Method

### 2.1. Datasets

We selected two datasets [42] for MI classification to validate our methods. First, we chose the BCI competition IV data set 1 as the training and test data set. This data set was provided by the BCI Research Institute in Berlin and contained two parts: the standard set and the evaluation set. The data of the four subjects (b, d, e, and g) were used for the analysis. The experimental process is shown in Figure 1. The sampling frequency of this experiment was 100 Hz, and each subject underwent 200 trials, resulting in 800 trials for the four subjects as the training and test data. We used EEG signals from three channels (C3, Cz, and C4).

The second dataset included the data from nine subjects from the BCI competition IV data set 2b. Three channels (C3, Cz, and C4) were used to record the EEG signals using a 250 Hz sampling rate. Each subject underwent 120 trials in 1–2 sessions and 160 trials in 3–5 sessions. We used five sessions for 720 × 9 trials for all subjects. The experimental process is shown in Figure 2.

The number of trials in each subject class was the same for both datasets. We filtered the 8–30 Hz signals using a Butterworth filter before analysis.

### 2.2. Preprocessing of the Raw Data

MI can cause ERD in the contralateral motor cortex and ERS in the ipsilateral cortex; these phenomena are reflected in changes in the energy of different frequency bands [43]. However, time-series signals cannot describe the features of these conditions. One promising method is a time–frequency transform, which expands the signal in two dimensions. A short-time Fourier transform (STFT) [44] is commonly used, in which a time-frequency localized window function is used for the transformation. The energy characteristics can be detected using a sliding window function that transforms the signals [45] because C3, C4, and Cz represent the dynamical change in the EEG of the MI [46]. Therefore, these three channels were used for the analysis.

As shown in Figure 3, the three channels were converted into a two-dimensional form and were mosaicked into an image using vertical stacking. For each image, the color depth indicates the signal energy of the different bands, the color change trend in the *x*-axis direction represents the time series, and the color change trend in the *y*-axis direction reflects the characteristics of the different frequency bands. STFT was applied to the time series for 4 s trials (during imagery period), with window sizes equal to 128 and 256 for the two datasets, respectively. Due to the difference in sampling rate, the sample sizes of the two datasets were 400 and 1000. Meanwhile, the frequency bands between 8 and 30 Hz were considered to represent motion-related bands. The process was repeated for three electrodes, which were C3, Cz, and C4. The results were vertically stacked in a way that the channel’s neighboring information was preserved. Finally, all spectrogram images were resized to 64 × 64 after the transformation for convenience and consistency in the subsequent calculations.

### 2.3. Different Data Augmentation Models

DA has been demonstrated to improve the performance of pattern recognition models in the computer vision field [47]. DA increases the complexity of the training model and reduces overfitting by adding artificial data. In this study, we compared the performance of different DA methods for MI classification using a DNN. In the following section, we briefly introduce the different data methods used in our research.

#### 2.3.1. Geometric Transformation (GT)

GT is an effective method that changes the geometry of the data. The method preserves the characteristics of the data and increases the diversity of the representation [48]. As shown in Figure 4 we used three GT methods for the DA of the MI signals:(1)Rotate the image 180° right or left on the *x*-axis (rotation);(2)Shift the images left, right, up, or down; the remaining space is filled with random noise (translation);(3)Perform augmentations in the color space (color-space transformation).

#### 2.3.2. Noise Addition (NA)

NA refers to the addition of random values to the raw data using a Gaussian distribution. Francisco et al. [49] demonstrated that NA significantly improves the performance and robustness of a model. A standard random uniform noise procedure was implemented to augment the raw data. The calculation is shown in the following equation:$$\tilde{x}=x+\mathrm{random}\left(-0.5,0.5\right)*\mathrm{noise}.$$

In our study, we randomly added Gaussian noise to the MI spectrogram data (Figure 5).

#### 2.3.3. Generative Model

Generative models use artificial data with features similar to that of the raw data; these models have a powerful feature mapping ability and provide a good representation of the original data. In this study, we evaluated the performance of three different generative models.

##### a. Autoencoder (AE)

A useful strategy for generative modeling involves an autoencoder (AE). As shown in Figure 6, an AE is a feed-forward neural network that is used for data dimensionality reduction, feature extraction, and model generation. The network contains two parts: the encoder z=f(x) is used to compress the input data, and the decoder r=g(z) restores the data that contains useful features.

##### b. Variational Autoencoder (VAE)

Variational autoencoders (VAEs) and AEs have a similar structure, but VAEs include constraints on the encoder to ensure that the output of the AE has a particular distribution and good robustness. A VAE can be defined as a directed model that uses learned approximate inferences [50]. To generate new data using a VAE, an encoder is used to obtain the hidden variable *z*, and the decoder then generates new data *x*. During training, the hidden variable learns the probability distribution from the input. In this study, we used the AE (Figure 6) and VAE (Figure 7) models described in Ref. [51].

##### c. Deep Convolutional Generative Adversarial Networks (DCGANs)

Another type of generative model for DA is a GAN. Goodfellow et al. originally proposed the GAN for data generation and conducted qualitative and quantitative evaluations of the GAN model by comparing it with deep learning networks and overlapping self-encoders [52]. A GAN uses the competition between two networks to achieve a dynamic balance to learn the statistical distribution of the target data. The generator first initializes a random noise vector $p_{z}$ and learns the distribution $P_{x}$ of the target parameter *X* by fitting a differentiable function to approximate $G\left(z;\theta_{G}\right)$. The discriminator uses the differentiable function approximator D() to predict the input variables from the actual target data distribution $P_{x}$ and not from the generated function. The optimization goal of the framework is to minimize the mean square error between the generated sample prediction label and the real sample label. The generator is trained to minimize the function $\log(1-D\left(G\left(z;\theta_{G}\right)\right)$. Hence, the optimization problem of the GAN can be defined as:$${\mathrm{Min}}_{G}{\mathrm{Max}}_{D}V\left(D,G\right)=E_{x\sim p\left(x\right)}\left[\log Dx;\theta_{G}\right]+E_{z\sim p\left(z\right)}\left[\log\left(1-Dx;\theta_{G}\right)\right],$$
where *V* represents the value function and *E* represents the expected value. *x* is the RD, *z* is the random noise vector, and *P*(·) is the distribution. The discriminator aims to distinguish whether the generated data are real or not. Thus, cross-entropy is adopted as the loss for this binary classification:$${\mathrm{Loss}}_{D}=-\frac{1}{N}\overset{N}{\underset{i=1}{\sum }}y_{i}\log\left(D\left(x_{i}\right)\right)-\frac{1}{N}\overset{N}{\underset{i=1}{\sum }}\left(1-y_{i}\right)\log\left(1-D\left(x_{i}\right)\right).$$

During the training of GANs, the objective is to find the Nash equilibrium of a non-convex game with continuous, high-dimensional parameters. GANs are typically trained using gradient descent techniques to determine the minimum value of a cost function. The GAN learns the feature representation without requiring a cost function, but this may result in instability during training, which often generates a meaningless output [53]. To address this problem, many researchers have proposed various morphing shapes. In the field of image processing, the DCGAN was proposed [54], and the authors focused on the topology of the DCGAN to ensure stability during training. The discriminator creates filters based on the CNN learning process and ensures that the filters learn useful features of the target image. This generator determines the feature quality of the generated image to ensure the diversity of the generated samples. Since the DCGAN shows excellent performance for image features in hidden space [55], we chose the DCGAN to generate the EEG images. The DCGAN differs from the GAN in the following model structure:The pooling layer is replaced by fractional-strided convolutions in the generator and by strided convolutions in the discriminator.Batch normalization is used in the generator and discriminator, and there is no fully connected layer.In the generator, all layers except for the output use the rectified linear unit (ReLU) as an activation function; the output layer use tanh.All layers use the leaky ReLU as the action function in the discriminator.

In this study, we referred to the structure of DCGAN in Cubuk et al. [48] and implemented it as a baseline; the generator and discriminator networks were extended to capture more relevant features from the MI-EEG datasets. The detail of the network structure is described in the following.

#### 2.3.4. Generator Model

Due to the weakness and non-stationary nature of the features, a generator is necessary to create high precision. To guarantee the performance of DA, the generator model should maintain a balanced condition between the discriminator and the generator. As shown in Figure 8, a six-layer network was proposed in our study.

A three-channel RGB spectrogram MI image was generated by a random vector using the generator. The operation of up-sampling and convolution guaranteed the output was consistent with the original training dataset. The number of channels of each deconvolution layer was halved, and the output tensor was doubled. Finally, the last generated image was output by the tanh activation layer. Details of the generator are summarized in Table 2.

#### 2.3.5. Discriminator Model

As shown in Figure 9, the discriminator network consisted of a deep convolution network that aimed to distinguish whether the generated image came from the training data or the generator. Details of the discriminator are summarized in Table 3.

“Adam” was used as the optimizer with the following parameters: learning rate = 2 × 10^−4^, batch size = 128, and training epoch = 20. For every subject in the two datasets, we used a 10-fold cross-validation to divide the data and train the network. The network structure of the DCGAN is shown in Figure 10.

### 2.4. Performance Verification of the Data Augmentation

It is well known that the clarity and diversity of the GD are important evaluation indicators. Researchers conducted a systematic review of the quality evaluation of the GD [56]. For image data, visualization is a reliable method because problems can be easily detected in the GD. However, this method does not provide quantitative indicators of the quality of the GD. The inception score is a commonly used quantitative index of the quality of GD. This method assesses the accuracy of the GD using an inception network. The FID is an improved version of the inception score and includes the probability distribution and a similarity measure between the GD and RD [53]. In this method, the features of the data are extracted using the inception network [57], and a Gaussian model is used to conduct spatial modeling of the features. The FID is calculated according to the mean value and covariance of the Gaussian model:$$\mathrm{FID}\left(r,g\right)={\left\|\mu_{r}-\mu_{g}\right\|}_{2}^{2}+\mathrm{Tr}\left(\Sigma_{r}+\Sigma_{g}-2{\left(\Sigma_{r}\Sigma_{g}\right)}^{2}\right),$$
where *r* represents the RD, *g* represents the GD, and Tr is the trace of the matrix. A small FID value indicates a high similarity between the GD and RD and a good DA performance. We compared the augmentation performance of the DCGAN with those of the GT, NA, and other generative models.

### 2.5. Evaluation of the MI Classification Performance after the Augmentation

It is expected that a good DA performance improves the performance of the classifier, especially for classification models based on a DNN, which is sensitive to the size of the dataset. CNNs are often used in image classification tasks and result in a good performance. CNNs often provide better performance than traditional methods for the processing of EEG signals [58,59,60].

A CNN is a multi-layered neural network consisting of a sequence of convolution, pooling, and fully connected layers. Each neuron is connected to the previous feature map by the convolution kernel. The convolution layer extracts the features of the input image using the kennel size, and the pooling layer is located between the continuous convolution layers to compress the data and parameters and reduce overfitting. More advanced features can be extracted with a larger number of layers. The fully connected layer transforms the output matrix from the last layer to an *n*-dimensional vector (*n* is the number of classes) to predict the distribution of the different classes. Backpropagation is utilized to decrease the classification error.

In the convolution layer, the input image can be convolved with a spatial filter to form the feature map and output function, which is expressed as:$$X_{j}^{l}=f\left(\underset{i\in M_{j}}{\sum }X_{i}^{l-1}\times w_{ij}^{l}+b_{j}^{l}\right).$$

This formula describes the *j*th feature map in layer *l*, where $X_{j}^{l}$ is calculated using the previous feature map $X_{i}^{l-1}$ multiplied by the convolution kernel $W_{ij}^{l}$ and adding a bias parameter $b_{j}^{l}$. Finally, the mapping is completed using the ReLU function f(a):$$f\left(a\right)=\mathrm{ReLU}\left(a\right)=ln\left(1+e^{a}\right).$$

The pooling layer is sandwiched in the continuous convolution layer to compress the amount of data and parameters and reduce overfitting. The max-pooling method was chosen in this work as follows:$$X_{j,k}^{l}=\max_{0\leq m,n\leq s}\left(X_{j\cdot s+m,k\cdot s+n}^{l-1}\right).$$
where *j* and *k* are the locations of the current feature map $X_{j}^{l}$ and *s* stands for pooling size. The double fully connected layer structure can effectively translate the multi-scale features of the image. Considering the multiple influencing factors of time, frequency, and channel, this study used double fully connected layers to improve the performance gain of the softmax layer. Two-way softmax in the last layer in the deep networks was used to predict the distribution of the two motor imagery tasks:$$y_{i}=\frac{\exp\left(\sum x_{i}\cdot w_{i,j}+b_{j}\right)}{\sum \exp\left(\sum x_{i}\cdot w_{i,j}+b_{j}\right)},$$
where $x_{i}$ is the *i*th feature map and $y_{i}$ represents an output probability distribution. The gradient of the backpropagation was calculated according to the cross-entropy loss function:$$\mathrm{Loss}=-\left[y\log\tilde{y}+\left(1-y\right)\log\left(1-\tilde{y}\right)\right].$$

Furthermore, we used the stochastic gradient descent (SGD) optimizer with a learning rate of 1 × 10^−4^ to improve the speed of the network training:$$W^{k}=W^{k}-\mu \frac{tialE}{tialW^{k}},$$
$$b_{k}=b_{k}-\mu \frac{tialE}{tialb_{k}},$$
where μ is the learning rate, $W^{k}$ represents the weight matrix for kernel *k*, and $b_{k}$ represents the bias value. E represents the difference between the desired output and the real output.

In our study, an eight-layer neural network structure was used to classify the two-class MI signals (Figure 11).

Considering the multiple influencing factors of time, frequency, and channel, we used two fully connected layers to improve the performance gain of the softmax layer [58]. The gradient of the backpropagation was calculated using the cross-entropy loss function, and we used a stochastic gradient descent with momentum (SGDM) optimizer with a learning rate of 1 × 10^−4^ to improve the speed of network training. To reduce computation time and prevent overfitting, we adopted the dropout operation. The parameters of the proposed CNN model are summarized in Table 4:

The average classification accuracy and kappa value were used as evaluation criteria to compare the performances of all methods. We divided the RD into training data and test data using 10-fold cross-validation [61]. In each dataset, 90% of the trials combined with the GD were selected randomly as the training set, and the remaining 10% of the RD was used as the test set. This operation was repeated 10 times.

The kappa value is a well-known method for evaluating EEG classifications because it removes the influence of random errors. It is calculated as:$$\mathrm{kappa}=\frac{\mathrm{accuracy}-\mathrm{random}}{\mathrm{random}}.$$

We determined the optimal ratio of the GD and RD by comparing the classification accuracies of different ratios of the GD and RD.

## 3. Experimental Results

### 3.1. Results of the Freéchet Inception Distances for Different Data Augmentation Methods

In this experiment, we used five DA methods to generate artificial MI-EEG data. We executed data augmentation based on a spectrogram MI signal (Section 2.2) for each subject independently. Furthermore, there were 200 trials for one subject in dataset 1 and 720 trials for one subject in dataset 2b. As for the GT and NA methods, all trials from one subject were randomly sampled for training. Meanwhile, the 10-fold cross-validation strategy was used to train the generated model for AE, VAE, and DCGAN. The quality of the GD was assessed using the FID, which is the probability distance between the two distributions. A lower value represents a better DA performance. As shown in Table 5, the data generated by the GT were considerably different from the RD. The quality of the data generated by the DCGAN was significantly higher than that of the other models, although the FID results were not ideal. Among the three DA methods based on generative models, the score of dataset 2b was better than that of dataset 1. Some possible explanations are listed in the following:One subject for each of 200 trials and 720 trials in datasets 1 and 2b, respectively. A larger-scale training data improved the robustness and generalization of the model.Due to the difference in sampling rate, the sample sizes of the two datasets were 400 and 1000 (datasets 1 and 2b, respectively). More samples would be helpful to improve the resolution of the spectrogram.During the experimental process, dataset 2b designed the cue-based screening paradigm that aimed to enhance the attention of the subjects before imagery. However, there was no similar set in dataset 1. This setting may lead to a more consistent feature distribution and higher quality for MI spectrogram data.

In summary, the sampling rate, design of the paradigm, and the dataset scale could obviously influence the quality of the generated data.

Figure 12a,b shows the analysis of variance (ANOVA) statistics of the different methods for the BCI Competition IV datasets 1 and 2b, respectively. There were statistically significant differences between the different DA methods.

To compare the effects of different DA methods, we show different generated spectrogram MI data in Figure 13.

### 3.2. Classification Performance of Different Data Augmentation Methods

We used the average classification accuracy and mean kappa value to evaluate both datasets. First, we determined the classification accuracies using DA. The results of the classification accuracy and standard deviation are shown in Table 6 and Table 7, and the kappa value results and standard deviations of the methods are presented in Table 4 and Table 5. The average classification accuracies of the CNN methods without DA were 74.5 ± 4.0% and 80.6 ± 3.2% for datasets 1 and 2b, respectively (baseline). The NA-CNN, VAE-CNN, and DCGAN-CNN provided higher accuracies than the baseline for both datasets (Table 2 and Table 3). The results of the different ratios of RD and GD indicated no positive correlation between the accuracy and the proportion of training data from the GD. In this study, the ratio of 1:3 (RD:GD) provided the optimal DA performance. The average classification accuracy of the CNN-DCGAN was 12.6% higher than the baseline for dataset 2b and 8.7% higher than the baseline for dataset 1. We also noticed that none of the ratios provided satisfactory results for the CNN-GT model. One possible explanation is that the rotation may have adversely affected the information in the EEG channel, resulting in incorrect labels.

The mean kappa value of the CNN-DCGAN was the highest among the methods, indicating that the DCGAN obtained sufficient knowledge of the features of the EEG spectrogram. As shown in Table 8 and Table 9, the performance of the three generative models was superior to that of the other DA methods. In addition, the standard deviation of the kappa value was relatively small, indicating the good stability and robustness of this method. Regardless of the RD:GD ratio, the results of the CNN-DCGAN showed a high degree of consistency for the average classification accuracy. Overall, the results demonstrated that this strategy provided the most stable and accurate classification performance.

ANOVA and paired *t*-tests were performed. We compared the CNN-DCGAN with other CNN-DA to determine the optimal DA method (with the optimal ratio) and compared the CNN-DCGAN with the CNN to verify the effectiveness of augmentation. Statistically significant differences were observed and are shown in Figure 14. DA using DCGAN effectively improved the performance of the classification model (CNN). Among the proposed CNN-DA methods, CNN-DCGAN outperformed in terms of the classification performance. In addition, the *p*-values for the comparison of the CNN-DCGAN and proposed methods are shown in Table 10. The classification performance of CNN-DCGAN was significantly higher than other methods (*p* < 0.01). Although CNN-VAE was second to CNN-DCGAN in dataset 2b (*p* < 0.05), CNN-DCGAN obtained the best *p*-values. In summary, the DCGAN provided effective DA and resulted in the highest classification performance.

### 3.3. Comparison with Existing Classification Methods

We compared the classification performance of the CNN-DCGAN hybrid model with that of existing methods (Figure 15). The results are shown in Table 11. The CNN-DCGAN exhibited a 0.072 improvement in the mean kappa value over the winning algorithm for the BCI competition IV dataset 2b [62]. The strategy proved favorable in the DNN for the classification of the MI-EEG signal, and the proposed model achieved comparable or better results than the other methods.

## 4. Discussion

In this study, we proposed a method to augment and generate EEG data to address the problem of small-scale datasets in deep learning applications for MI tasks. The BCI Competition IV dataset 1 and 2b were used to evaluate the method. We used a new form of input in the CNN that considered the time–frequency and energy characteristics of the MI signals to perform the classifications. Different DA methods were used for the MI classification. The results showed that the classification accuracy and mean kappa values of the DA based on DCGAN were highest for the two datasets, indicating that the CNN-DCGAN was the preferred method to classify MI signals and DCGAN was an effective DA strategy.

Recently, a growing number of researchers have used deep learning networks to decode EEG signals [60]. However, it remains a challenge to find the optimal representation of an EEG signal that is suitable for a classification model based on different BCI tasks. For example, the number of channels and the selection of frequency bands are crucial when choosing input data; therefore, different input parameters need to match the neural networks with different structures. Researchers require sufficient knowledge of the implications of using different EEG parameters and choosing classification networks for different forms of input data. In Vernon et al. [68], the deep separate CNN achieved better classification results for time-domain EEG signals because the model structure was highly suitable for the time-domain characteristics of the steady-state visually evoked potentials. AlexNet had excellent classification performance for time–frequency EEG signals after a continuous wavelet transform in Chaudary et al. [69]. In this study, we concluded that MI signals based on time–frequency representation was more suitable as the input of the DNN classification model. In future studies, we will investigate which useful features the convolution kernel learns from the EEG and optimize the structure and parameters of the model accordingly.

In applications of EEG decoding, the performance of a classification model based on DNNs is directly related to the scale of the training data. However, in a BCI system, it is difficult to collect large-scale data due to the strict requirements regarding the subject and experimental environment. Data augmentation provides an enlightening strategy to solve this limitation and we have verified its effectiveness in this manuscript. In a previous study, some research has shown that generative networks provided good performance for the deep interpretation of EEG signals [70]. Therefore, future studies could focus on generative networks to interpret the physiological meaning of EEG signals in depth to improve the explanation of EEG signals and investigate how to design a specific DA model with the requirements of specific tasks. Finally, by combining these methods, we hope to achieve accurate identification of MI tasks using a small sample size.

As an important technology focused on rehabilitation [71,72], MI-BCI aims to replace or recover the motor nervous system functionality that is lost due to disease or injury. In the application of DA for MI-EEG, future work could extend this work in clinical BCI tasks. For example, due to the cerebral injury of stroke patients, it is difficult to collect the available EEG signal that may lead to a long calibration cycle. One approach worth doing is to generate artificial data based on limited real data using a DA strategy and train the decoding model using these data. Additionally, we could also use the proposed methods to assess the difference between patients and healthy people, utilizing the generator to produce “healthy” EEG data based on patients and discriminator models to distinguish whether the current EEG signal is healthy or not. Based on DA for EEG, we may establish the correlation between the EEG signal with a rehabilitation condition. Rafael and Esther [73] used DA methods to simulate EMG signals with different tremor patterns for patients suffering from Parkinson’s disease and extended them to different sets of movement protocols. Furthermore, the proposed method has the potential to extend the application in rehabilitation and clinical operations based on BCI in practical applications.

## 5. Conclusions

In this study, we proposed a DA method based on the generative adversarial model to improve the classification performance in MI tasks. We utilized two datasets from the BCI competition IV to verify our method and evaluate the classification performance using statistical methods. The results showed that the DCGAN generated high-quality artificial EEG spectrogram data and was the optimal approach among the DA methods compared in this study. The hybrid structure of the CNN-DCGAN outperformed other methods reported in the literature in terms of the classification accuracy. Based on the experimental results, we can conclude that the proposed model was not limited by small-scale datasets and DA provided an effective strategy for EEG decoding based on deep learning. In the future, we will explore specific DA strategies for different mental tasks or signal types in a BCI system.

## Figures and Tables

**Figure 1 sensors-20-04485-f001:**
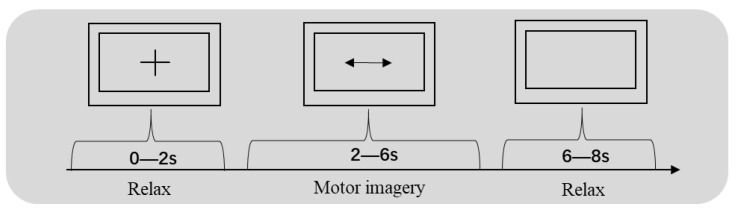
Schematic diagram of the experiment and the timing during a session using the Brain–Computer Interface (BCI) Competition IV dataset 1.

**Figure 2 sensors-20-04485-f002:**
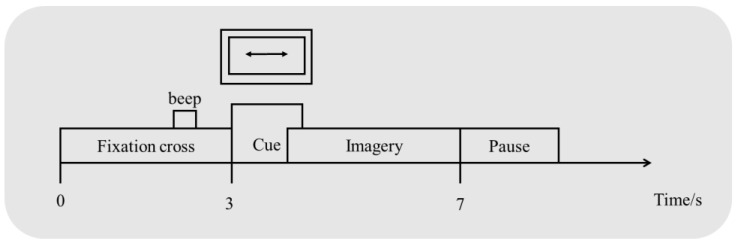
Schematic diagram of the experiment and the timing during a session using the BCI Competition IV dataset 2b.

**Figure 3 sensors-20-04485-f003:**
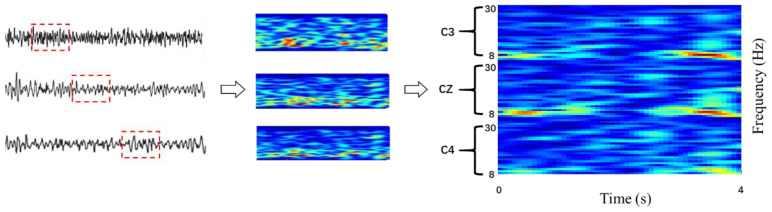
Input images with the electrodes after the short-time Fourier transform (STFT).

**Figure 4 sensors-20-04485-f004:**
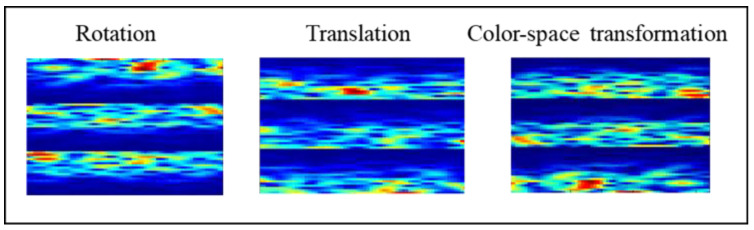
Data augmentation using geometric transformations.

**Figure 5 sensors-20-04485-f005:**
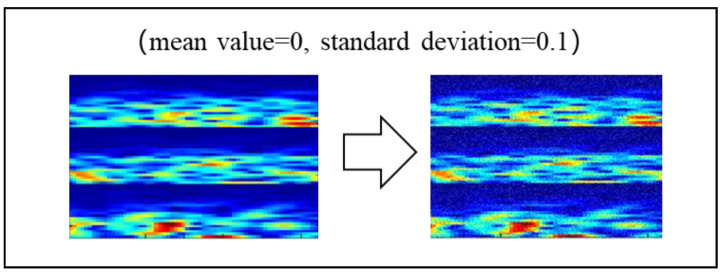
Data augmentation using noise addition.

**Figure 6 sensors-20-04485-f006:**
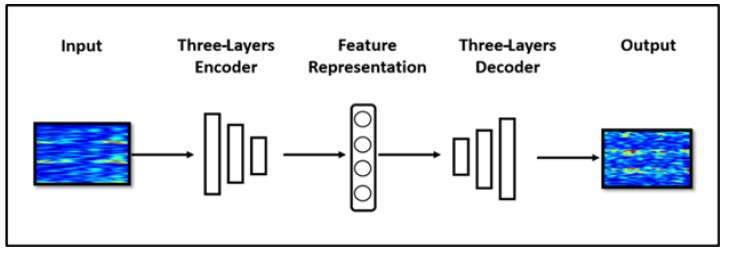
Data augmentation using an autoencoder.

**Figure 7 sensors-20-04485-f007:**
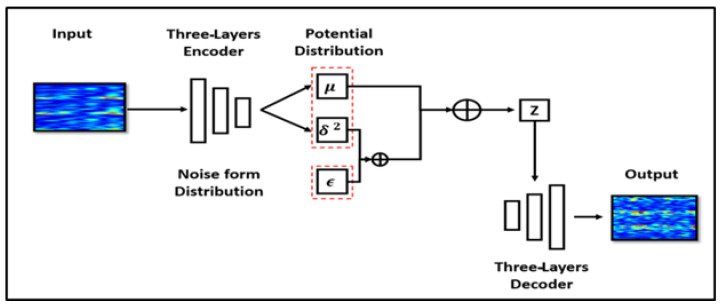
Data augmentation using a variational autoencoder.

**Figure 8 sensors-20-04485-f008:**
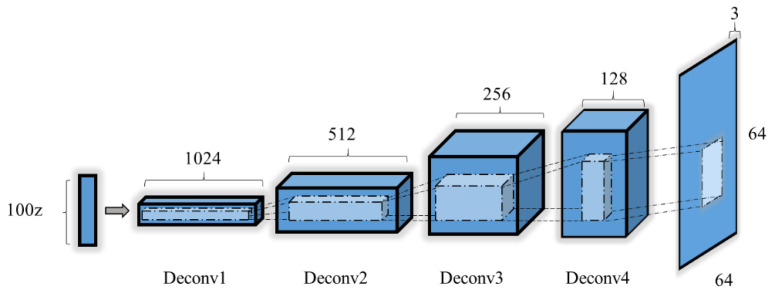
The structure of the generator.

**Figure 9 sensors-20-04485-f009:**
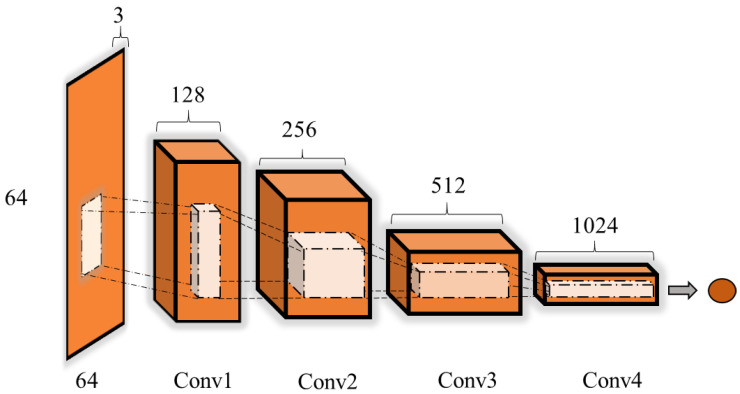
The structure of the discriminator.

**Figure 10 sensors-20-04485-f010:**
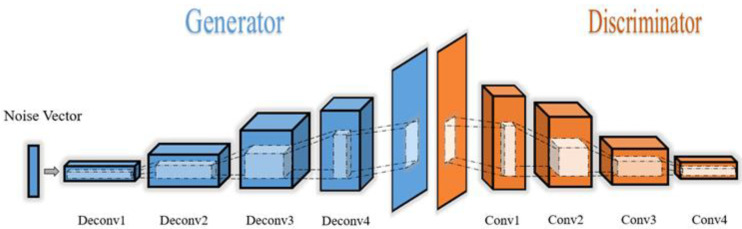
The structure of the deep convolutional generative adversarial network (DCGAN) model.

**Figure 11 sensors-20-04485-f011:**
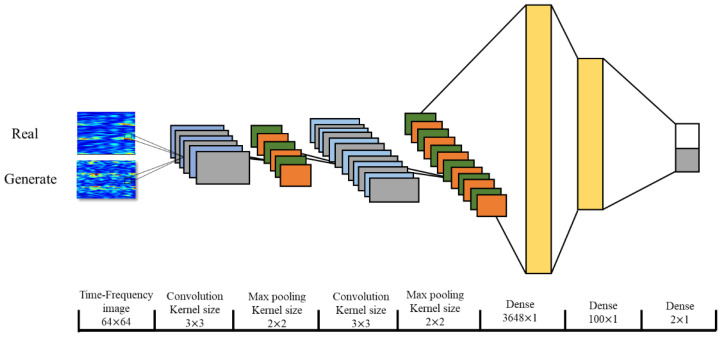
The structure of the convolutional neural network (CNN) model.

**Figure 12 sensors-20-04485-f012:**
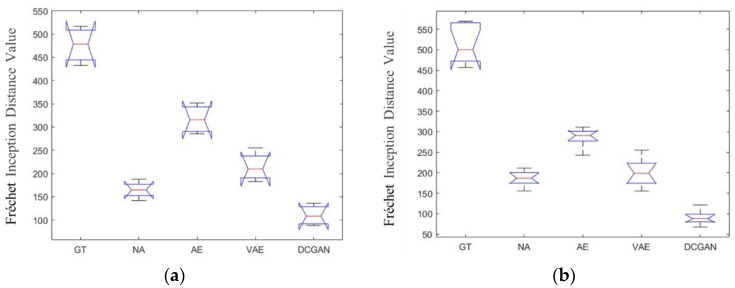
The ANOVA stats of the FID on data set 1 (**a**) and data set 2b (**b**).

**Figure 13 sensors-20-04485-f013:**
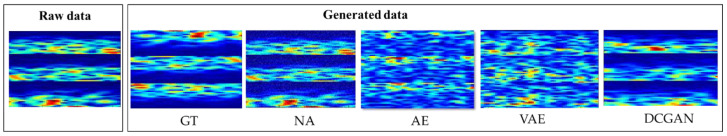
The post hoc data after data augmentation using different methods.

**Figure 14 sensors-20-04485-f014:**
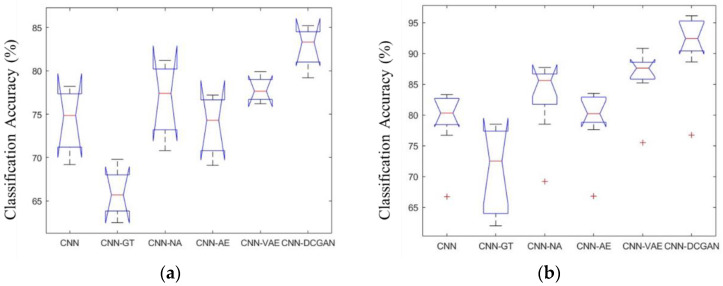
The ANOVA statistics of the classification accuracies of the methods for datasets 1 (**a**) and 2b (**b**).

**Figure 15 sensors-20-04485-f015:**
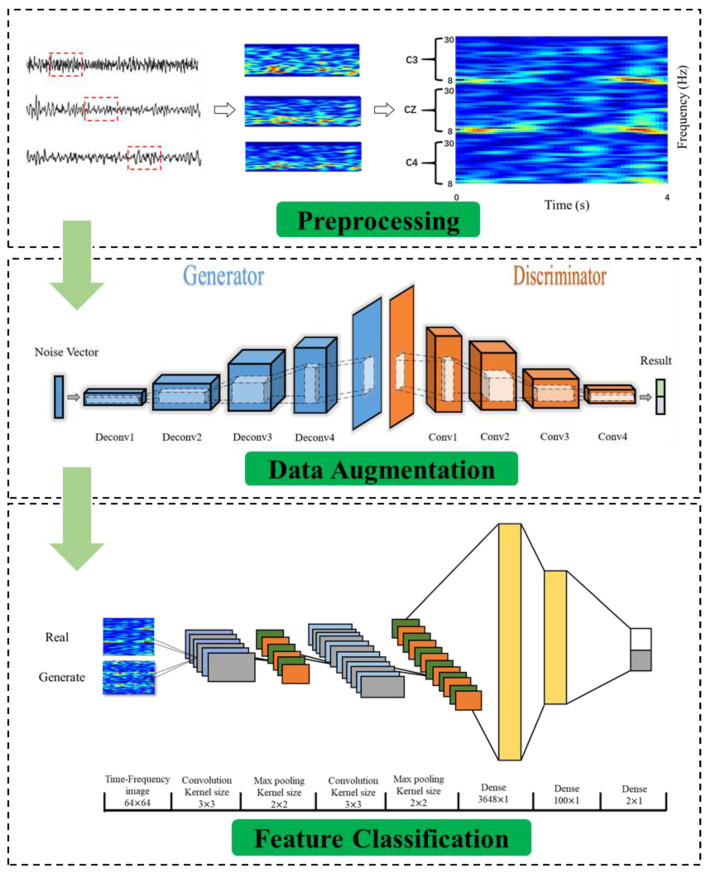
The hybrid network for MI classification.

**Table 1 sensors-20-04485-t001:** Data augmentation for motor imagery (MI).

Electroencephalogram (EEG) Pattern	Augmentation Methods	Limitations
Motor movement/imagery [25]	Recurrent generative adversarial network (GAN)	Shows good potential for time-series data generation but has limitations for image generation
Motor imagery [26]	Segmentation–recombination	Limited improvement for the diversity of feature distribution
Motor imagery [27]	Noise addition	May change and adversely affect the feature distribution
Motor imagery [28]	Empirical mode decomposition	Suitable for time-series data generation but has limitations for image generation;
Motor imagery [29]	GAN	instability during training may result in meaningless output
Motor imagery [30]	Geometric transformation and noise addition	Easy to obtain motion-related information after geometric transformation but limited improvement for the diversity of data generation
Motor imagery [31]	Sliding windows	Easy to lose motion-related information after changing the window size
Motor imagery [32]	Geometric transformation	Easy to lose motion-related information after the geometric transformation

**Table 2 sensors-20-04485-t002:** Detailed architecture for the generator.

Layers	Type	Filter Size	Output Dimension	Activation	Note
Input	1		(100,1,1)	ReLU	
Batch norm			(100,1,1)		Momentum = 0.8
Deconvolution	2	3 × 3 (1024)	(1024,4,4)	ReLU	
Batch norm			(1024,4,4)		
Deconvolution	3	3 × 3 (512)	(512,8,8)	ReLU	
Batch norm			(512,8,8)		
Deconvolution	4	3 × 3 (256)	(256,16,16)	ReLU	
Batch norm			(256,16,16)		
Deconvolution	5	3 × 3 (128)	(128,32,32)	ReLU	
Batch norm			(128,32,32)		
Output	6	3 × 3 (3)	(3,64,64)	Tanh	

ReLU: Rectified linear unit.

**Table 3 sensors-20-04485-t003:** Detailed architecture for the discriminator.

Layers	Type	Filter Size	Output Dimension	Activation	Note
Input			(3,64,64)		
Convolution	1	3 × 3	(128,32,32)	Leaky ReLU	Dropout rate = 0.25 Momentum = 0.8
Dropout			(128,32,32)		
Convolution	2	3 × 3	(256,16,16)	Leaky ReLU	
Dropout			(256,16,16)		
Batch norm			(256,16,16)		
Convolution	3	3 × 3	(512,8,8)	Leaky ReLU	
Dropout			(512,8,8)		
Batch norm			(512,8,8)		
Convolution	4	3 × 3	(1024,4,4)	Leaky ReLU	
Dropout			(1024,4,4)		
Flatten			(16384)		
Output	5		(1)	Sigmoid	

**Table 4 sensors-20-04485-t004:** Detailed architecture for the CNN.

Layers	Type	Filter Size	Stride	Output Dimension	Activation	Mode
Input	1			(64,64,3)		Valid
Convolution	2	3 × 3	(1,1)	(64,64,8)	ReLU	
Max-pooling	3	2 × 2		(32,32,8)		
Convolution	4	3 × 3		(32,32,8)		
Max-pooling	5	2 × 2		(16,16,8)		
Dense	6			(10,1)		
Dense	7			(2,1)	Softmax	

**Table 5 sensors-20-04485-t005:** Results for the difference Freéchet inception distances (FIDs).

Dataset	Mean Difference of the FID (Generated Value−Real Value)				
	GT	NA	AE	VAE	DCGAN
Dataset 1	487.7	159.1	323.5	277.1	126.4
Dataset 2b	501.8	188.5	273.6	203.4	98.2

GT: Geometric Transformation, NA: Noise Addition, AE: Autoencoder, VAE: Variational Autoencoder, DCGAN: Deep Convolutional Generative Adversarial Network.

**Table 6 sensors-20-04485-t006:** Classification accuracy of the methods for the BCI competition IV dataset 1 (baseline: 74.5 ± 4.0%).

	Ratio	Accuracy% (Mean ± std. dev.)				
Method		1:1	1:3	1:5	1:7	1:9
CNN-GT		70.5 ± 2.0	68.5 ± 3.2	69.7 ± 1.8	63.5 ± 2.1	68.5 ± 3.7
CNN-NA		76.5 ± 2.2	77.8 ± 3.5	72.1 ± 5.2	69.8 ± 3.5	70.3 ± 3.9
CNN-AE		75.6 ± 3.0	78.2 ± 1.8	77.6 ± 3.5	72.0 ± 3.7	68.2 ± 5.2
CNN-VAE		77.8 ± 3.4	78.2 ± 2.2	75.4 ± 3.6	73.1 ± 2.2	70.8 ± 3.9
CNN-DCGAN		82.5 ± 1.7	83.2 ± 3.5	80.9 ± 2.1	75.5 ± 4.6	78.6 ± 2.6

**Table 7 sensors-20-04485-t007:** Classification Accuracy of the methods for the BCI competition IV dataset 2b (baseline: 80.6 ± 3.2%).

	Ratio	Accuracy% (Mean ± std. dev.)				
Method		1:1	1:3	1:5	1:7	1:9
CNN-GT		70.8 ± 4.1	73.2 ± 2.1	57.2 ± 3.3	65.6 ± 2.2	59.7 ± 3.2
CNN-NA		82.3 ± 1.8	86.2 ± 3.1	81.3 ± 3.0	84.5 ± 4.1	84.3 ± 6.7
CNN-AE		80.3 ± 2.5	83.2 ± 3.1	78.6 ± 2.5	75.9 ± 2.1	85.3 ± 3.4
CNN-VAE		85.3 ± 5.3	87.6 ± 2.3	87.7 ± 3.6	86.1 ± 2.8	85.9 ± 2.7
CNN-DCGAN		89.5 ± 2.7	93.2 ± 2.8	91.8 ± 2.2	87.5 ± 3.5	86.6 ± 3.2

**Table 8 sensors-20-04485-t008:** Mean kappa values of the methods for the BCI competition IV dataset 1 (baseline: 0.4018 ± 0.048%).

	Ratio	Mean Kappa Value% (Mean ± std. dev.)				
Method		1:1	1:3	1:5	1:7	1:9
CNN-GT		0.3205 ± 0.037	0.3010 ± 0.058	0.3120 ± 0.048	0.2880 ± 0.075	0.3120 ± 0.025
CNN-NA		0.3678 ± 0.032	0.3775 ± 0.086	0.3420 ± 0.037	0.3088 ± 0.042	0.3189 ± 0.052
CNN-AE		0.3660 ± 0.075	0.3976 ± 0.057	0.3887 ± 0.056	0.3435 ± 0.057	0.3250 ± 0.021
CNN-VAE		0.4098 ± 0.018	0.4119 ± 0.022	0.3976 ± 0.057	0.3759 ± 0.017	0.3259 ± 0.027
CNN-DCGAN		0.4538 ± 0.033	0.4679 ± 0.050	0.4352 ± 0.032	0.4012 ± 0.028	0.4155 ± 0.035

**Table 9 sensors-20-04485-t009:** Mean kappa value of the methods for the BCI competition IV dataset 2b (baseline: 0.4789 ± 0.077%).

	Ratio	MEAN Kappa Value% (Mean ± std. dev.)				
Method		1:1	1:3	1:5	1:7	1:9
CNN-GT		0.332 ± 0.075	0.321 ± 0.066	0.227 ± 0.069	0.287 ± 0.067	0.235 ± 0.045
CNN-NA		0.468 ± 0.072	0.588 ± 0.054	0.539 ± 0.062	0.526 ± 0.035	0.591 ± 0.087
CNN-AE		0.498 ± 0.026	0.525 ± 0.071	0.496 ± 0.038	0.452 ± 0.056	0.578 ± 0.066
CNN-VAE		0.535 ± 0.087	0.591 ± 0.054	0.595 ± 0.028	0.578 ± 0.077	0.546 ± 0.089
CNN-DCGAN		0.622 ± 0.078	0.671 ± 0.067	0.631 ± 0.055	0.605 ± 0.075	0.580 ± 0.032

**Table 10 sensors-20-04485-t010:** Paired *t*-test results (*p*-values) between the CNN-DCGAN and other methods.

Dataset	CNN vs. CNN-DCGAN	CNN-GT vs. CNN-DCGAN	CNN-NA vs. CNN-DCGAN	CNN-AE vs. CNN-DCGAN	CNN-VAE vs. CNN-DCGAN
Dataset 1	2.4 × 10^−5^	5.1 × 10^−5^	3.1 × 10^−4^	5.6 × 10^−4^	0.8 × 10^−2^
Dataset 2b	3.5 × 10^−4^	6.2 × 10^−6^	8.2 × 10^−3^	1.4 × 10^−4^	2.1 × 10^−2^

**Table 11 sensors-20-04485-t011:** The classification performance of different methods for the BCI competition IV dataset 2b.

Method	Researcher	Classifier	Mean Kappa Value
CSCNN *	[63]	CNN	0.663
CNN-VAE	[64]	VAE	0.603
NCA * + DTCWT *	[65]	SVM	0.615
FBCSP *	[62]	NBPW *	0.599
RES * + FBCSP	[66]	SVM	0.643
HCRF *	[67]	HCRF	0.622
CNN-DCGAN	Our method	CNN	0.671

* Some acronyms are defined in the following: Continuous Small Convolutional Neural Network: CSCNN; Neighbourhood Component Analysis: NCA; Dual-Tree Complex Wavelet Transform: DTCWT; Restricted Exhaustive Search: RES; Filter Bank Common Spatial Pattern: FBCSP; Naïve Bayesian Parzen Window: NBPW; Hidden Conditional Random Fields: HCRF.

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
