# Peer review of "Data Augmentation for Motor Imagery Signal Classification Based on a Hybrid Neural Network"

_sensors, 2020, doi:10.3390/s20164485_

Round 1
Reviewer 1 Report
The STFT process in section 2.2 and the CNN network structure configuration in section 2.5 are not described in detail, making it difficult for the reader to reproduce your method.
The proposed model seems to lack innovation.
Author Response
Please see the attchment

Reviewer 2 Report
Starting from the view that brain-computer interface (BCI), motor imagery (MI) has been widely used in the fields of neurological rehabilitation and robot control, the authors compare different methods for feature extraction and classification based on MI signals, in particular they investigate the performance of different data augmentation (DA) methods for the classification of MI data using a DNN. The results show that the deep convolutional generative adversarial network (DCGAN) provides better augmentation performance that the geometric transformation (GT), autoencoder (AE), and variational autoencoder (VAE). These findings are relevant implication for the applied context of motor imagery-BCI.
I have some comments that could help to improve the manuscript.
Point1.
The authors should provide more information about the experimental procedure used to produce the two datasets. Indeed, from Figures 1 and 2 it is clear that the experiment procedure and the timing during a session are different for the two sets. Indeed, results of the ANOVA showed that among the three DA methods based on generative models, the score of dataset 2b was better than that of dataset 1. The authors suggested that “One possible explanation is that the larger amount of training data improved the robustness of the generative model.” However, literature on cognitive and neural bases of motor imagery, as well as data on applied motor-BCI reveal that large differences can be found between findings depending on the kind of the experimental procedure used during training. Although the present study is a methodological one, however it looks at the best solution to be applied in a translational context. For this reason, it could be relevant trying to better tie the experimental procedure for BCI training used in the two dataset with the methodological part of the study.
Point2.
Please provide the label of the ordinate axis of Figure 7
Point 3.
Page 9, lines 284-285, ”There were statistically significant differences between the different DA methods”. The authors should provide post-hoc data showing the significant differences between DA methods
Point 4.
Page 11, lines 319-320, “ANOVA and paired t-tests were performed. We compared the CNN with the CNN-DA (optimal ratio) to determine the significance level of the results”. It is not clear to the reader what the authors are comparing. The standard deviation of the kappa values? Please clarify.
Point 5.
Page 11, lines 323-324, “The CNN-DCGAN obtained the best p-values for the two datasets (p<0.01), and the CNN-VAE had similar results for dataset 2b (p<0.01). In summary, the DCGAN provided effective DA and resulted in the highest classification performance.” However, when comparing different condition by an ANOVA or a t-test, to decide which condition produce the best effect, it is not correct to look at the p-values but one should look at the effect sizes. The authors should provide this information and use it to decide about which transformation produced the best results, of course with a word of caution in the case p-values indicate that all conditions are significant different with each other. Again, by inspecting Table 6, it is clear that a different pattern of data was obtained for the two datasets. Please pay attention to the comment above.
Point 6.
Page 13, lines 377-378, “Furthermore, the proposed method can be used for rehabilitation, clinical operations, and brain-machine interactive tasks in practical applications.” Following the above comments, the Referee believe that a deeper discussion needs to be made on the translational implications of the present study, better tying the present results with literature on applied motor imagery-BCI.
Author Response
Please see the attchment

Reviewer 3 Report
The manuscript describes a data augmentation for motor imagery EEG classification. It presents an extensive performance test result.
A few major comments.
1. For performance evaluation, the authors chose two data sets, both with only three EEG channels. It is important to evaluate the methods using EEG of more channels -- preferably full-scalp.
2. The theoretical part is seriously lacking. Starting from the introduction section. The paragraph from line 82 provides some brief, yet non-substantiated arguments against existing methods. There follows no definitive description of a technical gap. The manuscript thus fails to elaborate the concept/theory behind the proposed DCGAN in relation to technical gap.
3. Description of the key technical point of DCGAN is poorly organized nor detailed (merely two sections from line 186 through 220.) It is thus difficult to appreciate the technical merits in the design of the method.
4. Above all, there is an insignificant increase in classification performance by the proposed method compared with the CSCNN. The authors need to discuss how that indicates the potential value of DA.
Author Response
Please see the attchment
